# Carbon Nanotube Far Infrared Detectors with High Responsivity and Superior Polarization Selectivity Based on Engineered Optical Antennas

**DOI:** 10.3390/s21155221

**Published:** 2021-07-31

**Authors:** Xiansong Ren, Zhaoyu Ji, Binkai Chen, Jing Zhou, Zeshi Chu, Xiaoshuang Chen

**Affiliations:** State Key Laboratory of Infrared Physics, Shanghai Institute of Technical Physics, Chinese Academy of Sciences, Shanghai 200083, China; renxs@bupt.edu.cn (X.R.); jizhaoyu@outlook.com (Z.J.); 13120666402@163.com (B.C.); chuzeshi@mail.sitp.ac.cn (Z.C.)

**Keywords:** carbon nanotube far infrared detectors, optical antenna engineering, responsivity enhancement, polarization sensitivity enhancement, analysis of coupled light and thermal fields

## Abstract

Single-wall carbon nanotube (SWCNT) thin films are promising for sensitive uncooled infrared detection based on the photothermoelectric effect. The SWCNT film is usually shaped into a belt and diversely doped to form a p-n junction at the center. Under the illumination of a focused incident light, the temperature gradient from the junction to the contacts leads to photoresponse. When the SWCNTs are aligned in one direction, the photoresponse becomes polarization selective. Although a typical bowtie antenna can improve the responsivity and polarization extinction ratio by deep-subwavelength light focusing, the absolute absorptance of the junction region is only 0.6%. In this work, the antenna was engineered for a higher light coupling efficiency. By integrating a bottom metal plane at a specific distance from the SWCNT film and optimizing the antenna geometries, we achieved ultra-efficient impedance matching between the antenna and the SWCNTs, thus the absorptance of the junction region was further enhanced by 21.3 times and reached 13.5%, which is more than 3 orders of magnitude higher than that of the device without the engineered antenna. The peak responsivity was further enhanced by 19.9 times and responsivity reached 1500 V/W at 1 THz. The resonant frequency can be tuned by changing the size of the antenna. Over the frequency range of 0.5 THz to 1.5 THz, the peak responsivity was further enhanced by 8.1 to 19.9 times, and the polarization extinction ratio was enhanced by 2.7 to 22.3 times. The highest polarization extinction ratio reached 3.04 × 10^5^ at 0.5 THz. The results are based on the numerical simulations of the light and the thermal fields.

## 1. Introduction

Far infrared (including THz) detectors are important to a wide range of applications in the fields of astronomy, sensing, spectroscopy, imaging, defense and communications [1,2,3]. Since the photon energy in this regime is quite low, photonic detectors require cryogenic cooling systems, which are expensive and bulky, to maintain the performance. In this respect, thermal detectors have an advantage as they can operate in an uncooled mode, although they are generally less sensitive and slower than photonic detectors [4]. In this situation, new materials with a prominent thermoelectric effect, low specific heat capacity, and high thermal stability become promising candidates for future infrared thermal detectors [5,6,7,8,9,10,11,12]. Among those materials, single-wall carbon nanotubes (SWCNT) have attracted great attention due to their unique electrical, optical and thermal properties [4,8,9,10,11,12,13,14]. In particular, SWCNT thin films have been proposed and demonstrated as a potential infrared detection material by a lot of researchers, since they can be made highly uniform and can be diversely doped to form p-n junctions [8,13,14]. In addition, they are compatible with a variety of substrates. For a SWCNT thin film with a p-n junction, the far infrared illumination at the junction area will induce a self-driven photoresponse proportional to the difference between the Seebeck coefficients of the p- and n-doped parts and also proportional to the temperature rise at the junction, according to the PTE effect. When the assembled SWCNTs are aligned perpendicular to the electronic transportation direction, the reduced thermal conductivity benefits the local temperature rise at the junction area and then enhances the responsivity [9,13,14]. As a positive side effect, the alignment of the SWCNTs with intrinsic anisotropy leads to polarization selectivity, which is essential for infrared polarimetry.

Optical antennas have been widely employed to focus electromagnetic radiation into an area much smaller than the diffraction limit [15,16,17] and have been demonstrated to be able to enhance the responsivity of the aligned SWCNT film infrared detectors by 1 to 2 orders of magnitude and the polarization extinction ratio by 2 order of magnitudes [18]. Although it greatly improves the performance of SWCNTs-type infrared detectors, the absolute absorptance of the junction region (a 100 μm × 5 μm × 2 μm region at the p-n junction) is only 0.6%. In this work, we propose to significantly enhance the absorptance of the junction region (deep-subwavelength) in an aligned SWCNT film by engineering the optical antenna. A metal plane is added to induce a mirror image of the induced charges in the antenna. By setting the metal plane at a specific distance from the antenna, the radiation from the mirror image can interfere with that from the antenna in a proper manner, leading to a controlled radiation quality factor (*Q*_e_) that matches the absorption quality factor (*Q*_r_) for an optimized coupling efficiency. In another parameter dimension, the extension bar at each feeding point tips of the antenna is engineered to match the load resistance (*R*_load_) with the radiation resistance (*R*_rad_). When *R*_load_ equals to *R*_rad_, the antenna exhibits a maximum coupling efficiency. By engineering the light coupling properties of the antenna, the absorptance of the junction region is further increased by 21.3 times and reaches 13.5%. This absorptance is more than 3 orders of magnitude higher than that in the absence of the engineered antenna. Based on the coupled simulation of the light field and the thermal field, the temperature rise at the junction is further enhanced by 19.9 times compared to the case of an ordinary antenna integrated SWCNT film, and the responsivity (*R*_V_) reaches 1500 V/W at 1 THz. *R*_V_ is defined as the photovoltage divided by the incident optical power on the 100 μm × 5 μm area at the junction. The resonant frequency can be tuned by changing the size of the antenna. Over the range of 0.5 to 1.5 THz, the engineered antenna enhances the peak responsivities further by 8.1 to 19.9 times and enhances the peak polarization extinction ratios by 2.7 to 22.3 times, compared with the ordinary bowtie antenna [18]. It is worth noting that the peak polarization extinction ratio is higher than 1481 over the range from 0.5 to 1.5 THz and reaches an extremely high value of 3.04 × 10^5^ at 0.5 THz.

## 2. Materials and Methods

### 2.1. Aligned SWCNT Thin Film as an Effective Uniaxial Medium

A typical aligned SWCNT thin film infrared detector is shown in Figure 1a. The SWCNT film is considered as an effective uniaxial medium [18,19] because the diameter of each SWCNT and the inter-distances between them are within the deep subwavelength scale. The effective permittivity tensor is a diagonal matrix, as shown below [18,19]:(1)$$\varepsilon_{x}=\varepsilon_{z}=\varepsilon_{⊥}=\varepsilon_{e}+f\varepsilon_{e}\frac{\varepsilon_{\mathrm{SWCNT}}-\varepsilon_{e}}{\varepsilon_{e}+\frac{\left(1-f\right)\left(\varepsilon_{\mathrm{SWCNT}}-\varepsilon_{e}\right)}{2}}$$
(2)$$\varepsilon_{y}=\varepsilon_{∥}=\varepsilon_{e}+f\varepsilon_{e}\left(\varepsilon_{\mathrm{SWCNT}}-\varepsilon_{e}\right)$$

*ε*_⊥_ denote the relative permittivity of an aligned SWCNT film for the light field polarized perpendicular to the SWCNTs, and ε_‖_ denote the relative permittivity for the light field parallel to the SWCNTs. *f* represents the fill factor,
$\varepsilon_{\text{SWCNT}}$
is the permittivity of a single SWCNT, which can be approximated as a solid rod [20,21], and *ε*_e_ is the permittivity of the environment, which is air in this case. Based on the density of a SWCNT film [22,23], we assume *f* = 0.3. Since the SWCNT film contains both semiconductor nanotubes and metal nanotubes, 
$\varepsilon_{\text{CNT}}$
represents the average optical response. Based on the actual measurement results of SWCNT film under far-infrared conditions [24,25],
$\varepsilon_{\text{CNT}}$
is obtained through effective medium theory and modeled in the Drude-Lorentz form [20]. Based on this, we obtained the permittivity 
$\varepsilon_{⊥}$
and
$\varepsilon_{∥}$
of SWCNT.

### 2.2. Responsivity Obtained by Optical and Thermal Field Simulation

As shown in Figure 1a, the aligned SWCNT film (1 mm long, 5 μm wide, and 2 μm thick) is supported by a transparent dielectric substrate (teflon or SU8), and connected by two metal contacts. By diversely doping the two halves of the SWCNT film, a p-n junction is formed at the center. The Seebeck coefficient is positive in the p-doped region and negative in the n-doped region. In this way, when the incident light is focused at the p-n junction, a distributed temperature rise is created. The temperature at the junction is the highest and gradually decreases from the center to the two ends. Then, a photoresponse is created due to the photothermoelectric effect. The SWCNT film can be formed by first growing vertically and rolling down to form a macroscopic film [13,14,22], and then patterning the film by photolithography and etching [26,27]. The photoresponse can be estimated by numerical simulations of the optical field and the thermal field. The light field and the thermal field are described by the following equations (Equations (1)–(3)):(3)$${𝛻}^{2}E-k_{0}^{2}{\overset{↔}{\varepsilon }}_{r}E=0,$$
(4)$$-𝛻\cdot \left(\overset{↔}{k}{𝛻}^{2}T\right)=Q_{e}$$
(5)$$Q_{e}=\frac{1}{2}\varepsilon_{0}\omega \left(E\cdot \left(Im\left({\overset{↔}{\varepsilon }}_{r}\right)\cdot E\right)\right).$$
where *k*_0_ represents the wave vector in vacuum,
${\overset{↔}{\varepsilon }}_{\text{r}}$
is the relative permittivity tensor of the aligned SWCNT film, 
$\overset{↔}{k}$
is the thermal conductivity tensor of the aligned SWCNT film.
$\varepsilon_{0}$
is the vacuum permittivity, and *Q*_e_ denotes the light power absorption density. The absorbed light power becomes heat, so *Q*_e_ is the distributed heat source term in Equation (4). When a far infrared Gaussian beam (waist around 800 μm) is incident on the aligned SWCNT film (1 mm long, 5 μm wide, and 2 μm thick, as shown in Figure 1a), the light absorption distribution and the temperature rise distribution are simulated as the plots in Figure 1b,c. The frequency of the incident light is 1 THz and the beam waist is 400 μm. The simulations are based on the finite element method and through the software COMSOL. After the temperature distribution over the SWCNT film is worked out, the photovoltage
(ΔV)
due to the PTE effect (
$\text{Δ}V=\left(S_{p}-S_{n}\right)\text{Δ}T$)
is obtained. *S*_p_ and *S*_n_ denote the Seebeck coefficients of the p-doped and the n-doped regions, respectively. The details of simulation settings can be found in a previous study [18]. Since the real part of the parallel permittivity of the aligned SWCNT film (Re(
$\varepsilon_{∥}$
)) is negative in the frequency range from 0 to 2.8 THz [18], the film with a limited width (i.e., the length of the SWCNTs) can intrinsically support a localized surface plasmon (LSPR) mode. The LSPR frequency depends on the *W*_film_. Although the LSPR of the aligned SWCNT film could also enhance the light absorption, the light coupling efficiency is low and the light field is not concentrated at the junction region. Thus, the responsivity enhancement and the polarization extinction ratio are 1 to 2 orders of magnitudes smaller than those induced by an optical antenna [18].

## 3. Results

In order to increase the responsivity and the polarization extinction ratio of the aligned SWCNT film based detector, it is natural to add an antenna to concentrate the incident light at the junction region of the SWCNT film, as shown in Figure 2a. The SWCNT film is 1 mm long, 5 μm wide, and 2 μm thick, as shown in Figure 1a. The antenna has a bowtie shape. The length *L*_a_ and the width *W*_a_ are both 150 μm. The feed point overlaps the junction region. There is an extension bar (20 μm long and 2 μm wide) at the tip of each wing. It is also 2 μm away from the SWCNT film. Although an ordinary bowtie antenna can enhance the local *Q*_e_ at the junction by 240 times and thus enhance the responsivity by 23 times [18], the absolute absorptance of the junction region (*A*_j_) is only 0.6%. In order to further enhance *A*_j_ and the responsivity, we propose to integrate a bottom metal plane at a specific distance from the SWCNT film and the antenna, as shown in Figure 2b. As a result, the *Q*_e_ at the junction is further increased by 18 times (Figure 2c–e), and Δ*T* is further enhanced by 17.5 times (Figure 2f). *A*_j_ is enhanced from 0.6% to 11.7%, which is more than 3 orders of magnitude higher than that of the device without any antenna. The antenna dimensions *W_a_* and *L_a_* are adjusted from 150 μm to 171 μm after the bottom metal plane is integrated to keep the resonant frequency at 1 THz. When the thickness of the dielectric layer is as small as 13 μm, each wing of the bowtie antenna couples with its image due to the deep subwavelength thickness of the dielectric spacer and then forms a magnetic dipole, as exhibited in Figure 2c,e. At the magnetic dipole resonance, the incident light is efficiently coupled into the system, leading to an enhanced local field that significantly improves the responsivity of the carbon nanotube detector. The resonant frequency of the engineered bowtie antenna can be estimated by the antenna dimensions based on a patch antenna model [28]. The performance enhancement by the antenna engineering is attributed to the improved light coupling efficiency.

The local field intensity at the junction region can be derived from the coupled mode theory as [29]
(6)$$\frac{{\left|E_{\mathrm{loc}}\right|}^{2}}{{\left|E_{0}\right|}^{2}}=\frac{2A_{c}\lambda_{\mathrm{res}}}{\pi }\frac{Q}{Q_{\mathrm{rad}}}\frac{Q}{V_{\mathrm{eff}}}$$

*E*_loc_ is the amplitude of the local field, *E*_0_ the amplitude of the incident light, *A*_c_ is the effective aperture, *λ*_res_ the resonant wavelength, *V*_eff_ the effective mode volume. *Q* as the total quality factor is related to the radiation quality factor (*Q*_rad_) and the absorption quality factor (*Q*_abs_) through the equation *Q*^−1^ = *Q*_rad_^−1^ + *Q*_abs_^−1^. Based on the derivative of |*E*_loc_|^2^/|*E*_0_|^2^ with respect to *Q*_rad_, it is revealed that |*E*_loc_|^2^/|*E*_0_|^2^ reaches a maxima when *Q*_rad_ = *Q*_abs_, which is called a critical coupling condition or an impedance matching condition. When the system reaches a critical coupling status, the light coupling efficiency becomes a maxima. *Q*_abs_ is decided by the absorptance of the SWCNT film and the metal. In the absence of the bottom metal plane, the light mode is not very well confined (Figure 3a,b), and *Q*_rad_ is much smaller than *Q*_abs_, so the system is far from a critical coupling status. In the presence of the bottom metal plane, the radiation from the antenna and that from the mirror image of the induced charges will constructively or destructively interfere with each other as the thickness of the dielectric layer varies (Figure 3c,d), so *Q*_rad_ is tuned and the light coupling efficiency changes accordingly (Figure 3e). At a specific dielectric thickness, *Q*_rad_ matches *Q*_abs_ so the system reaches the critical coupling status and the light coupling efficiency is maximized. As revealed in Figure 3b,d, the local field induced by the optimized metal plane integrated antenna is several tens of times higher than that induced by the ordinary antenna.

The length of the extension bar at each tip of the two wings of the bowtie antenna is an important parameter that has a big impact on the coupling efficiency. As shown in Figure 4a, *A*_j_ varies non-monotonously with the length of the extension bar (*L*_bar_). The highest junction region absorptance occurs at *L*_bar_ = 40 μm. Correspondingly, the *∆**T* at the junction and the responsivity of the device also reaches the maxima at *L*_bar_ = 40 μm (Figure 4b, red line). This effect is attributed to the fine tuning of the antenna efficiency. According to the antenna theory [30], the antenna efficiency gets maximized when the radiation resistance (*R*_rad_) equals to the load resistance (*R*_load_). Concerning the bowtie antenna without the bottom metal plane, *R*_rad_ is around 73 Ω [30]. *R*_load_ is decided by the portion of the SWCNT film within the feed point region (defined by *L*_bar_ and marked out by the purple frame in Figure 4a). As *L*_bar_ increases, there are more and more SWCNTs bridging the feed point region in parallel, so *R*_load_ decreases. *R*_load_ = (1/σ)(1/*L*_bar_) can be calculated through the conductivity of the SWCNT film (σ = iωε). ω is the angular frequency, ε is the dielectric constant. Based on this expression, *R*_load_ is calculated to be 158 Ω at *L*_bar_ = 20 μm, and 78 Ω at *L*_bar_ = 60 μm. *R*_load_ is close to *R*_rad_ at *L*_bar_ = 60 μm, so the antenna efficiency and thus the junction region absorptance reach a maxima in this condition (Figure 4a black line). When the antenna is integrated with a metal plane, the length of the bowtie antenna (*L*_a_) needs to be enlarged to maintain the resonant wavelength. Since *R*_load_ increases with *L*_a_, the situation *R*_rad_ = *R*_load_ occurs at a smaller *L*_bar_ for the metal plane integrated antenna. As confirmed by the red line in Figure 4b,c, the optimized *A*_j_ appears around *L*_bar_ = 40 μm. In this case, the engineered antenna with *L*_bar_ = 40 μm enhances *A*_j_ by 21.3 times and enhances Δ*T* (*R*_V_) further by 19.9 times in comparison with the case of the ordinary bowtie antenna integrated device.

The spectra of *A*_j_, *∆**T*, and *R*_V_ for the SWCNT infrared detector integrated with the ordinary antenna and that with the engineered antenna are presented in Figure 5a,b. As confirmed by the spectra in Figure 5a,b, the engineered antenna enhances the peak absorptance and *∆**T* (*R*_V_) significantly at the resonant frequency for *y*-polarized incident light. The peak *R*_V_ reaches 1500 V/W, which is 19.9 times higher than that achieved by an ordinary antenna. For the *x*-polariztion, the engineered antenna induces an *R*_V_ of 0.045 V/W at 1 THz and the ordinary antenna induces an *R*_V_ of 0.043 V/W. Therefore, the engineered antenna enhances the PER by 19.1 times to 33,440. By altering the dimension of the antenna, the peak responsivity can be set at different frequencies. As shown in Figure 5c, as the antenna dimension rising from 95 μm to 379 μm, the peak responsivity shifts from 1.5 THz to 0.5 THz in the frequency domain. Concerning the antenna as large as 379 μm, it also supports higher order modes in this frequency range, as shown by the pink line in Figure 5c. The peak *R*_v_, peak PER, and the enhancement times of these two quantities compared to that of the ordinary antenna integrated devices are presented in Table 1.

## 4. Conclusions

In conclusion, we numerically demonstrated that light coupling engineering of an optical antenna integrated aligned SWCNT film based far infrared detector can further significantly enhance the performance. By integrating a bottom metal plane at a specific distance from the antenna and tuning the length of the extension bar at the tip of each wing of the antenna, the peak responsivity is further enhanced by 8.1 to 19.9 times over the frequency range from 0.5 THz to 1.5 THz, and the polarization extinction ratio is enhanced by 2.7 to 22.3 times. The highest peak responsivity in this range reaches 1500 V/W at 1 THz, 19.9 times higher than that induced by the ordinary antenna. The peak polarization extinction ratio is higher than 1481 over the range from 0.5 to 1.5 THz and reaches an extremely high value of 3.04 × 10^5^ at 0.5 THz. The mechanism of the enhancement is attributed to critical coupling and impedance matching, and this principle can be applied to a variety of systems to improve the light-matter interaction in subwavelength scales.

## Figures and Tables

**Figure 1 sensors-21-05221-f001:**
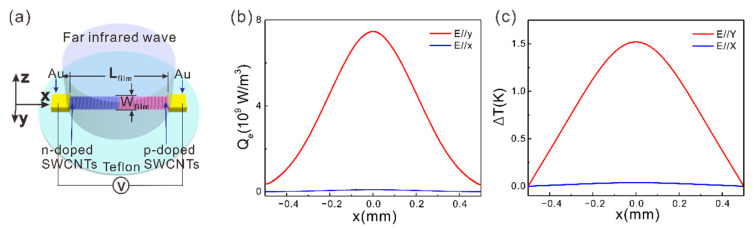
(**a**) Schematic of the aligned SWCNT film based far-infrared photodetector. *L*_film_ = 1 mm, *W*_film_ = 5 μm. (**b**) Light power absorption density in the aligned SWCNT film (*Q*_e_) versus *x* at *y* = 0 μm and *z* = 1 μm for *y*-polarized incident light (Red line) and that for *x*-polarized incident light (Blue line). (**c**) Δ*T* versus *x* at *y* = 0 μm and *z* = 1 μm for *y*-polarized incident light (Red line) and that for *x*-polarized incident light (Blue line).

**Figure 2 sensors-21-05221-f002:**
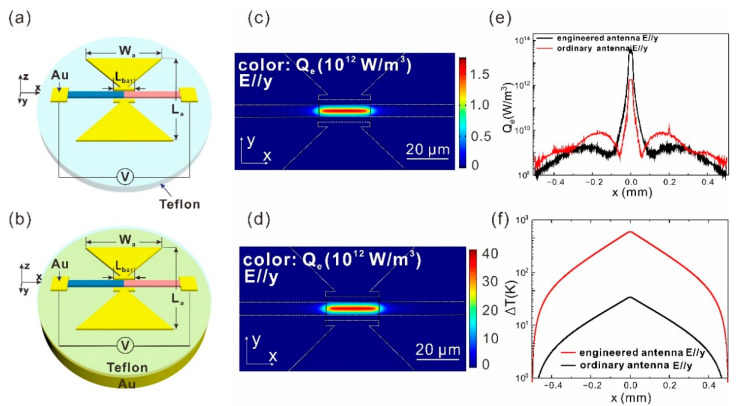
(**a**) Schematic of the ordinary antenna integrated aligned SWCNT film. *L*_a_ = *W*_a_ = 150 μm, *L*_bar_ = 20 μm. (**b**) Schematic of the engineered antenna integrated aligned SWCNT film. A gold plane is added under the teflon layer. *L*_a_ = *W*_a_ = 171 μm, *L*_bar_ = 20 μm. (**c**) Distribution of *Q*_e_ on the *x*-*y* section at *z* = 1 μm, cutting through the center of the ordinary antenna integrated aligned SWCNT film. (**d**) Distribution of *Q*_e_ in the same way as (**c**) for the engineered antenna integrated aligned SWCNT film. (**e**) *Q*_e_ varying along the *x*-axis at *y* = 0 μm and *z* = 1 μm through the center of the SWCNT film for the ordinary antenna integrated device (red line) and the engineered antenna integrated device (black line). (**f**) Δ*T* varying along the *x*-axis at *y* = 0 μm and *z* = 1 μm through the center of the SWCNT film for the ordinary antenna integrated device (red line) and the engineered antenna integrated device (black line). In this figure, the incident light is polarized in the *y*-direction.

**Figure 3 sensors-21-05221-f003:**
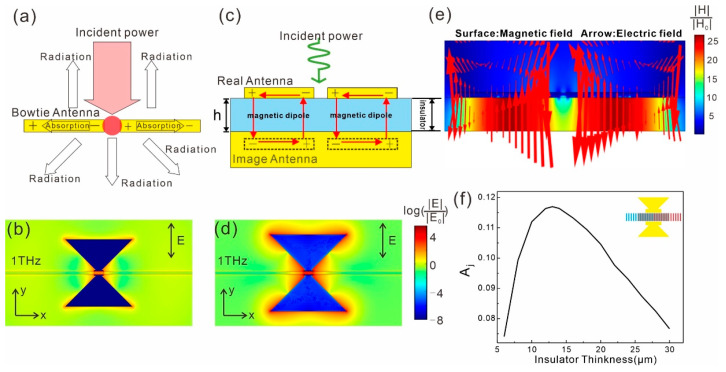
(**a**) Schematic diagram of modeling a bowtie antenna as a resonator using coupled mode theory. (**b**) Electric field enhancement distributions (log(|*E*|/|*E*_0_|)) on the *x*-*y* cross section at the center of the SWCNT film (*z* = 1 μm) integrated with the bowtie antenna at 1 THz for *y*-polarization. (**c**) Schematic diagram of the engineered antenna integrated SWCNT film. Each wing couples with its image and induces a magnetic dipole. (**d**) Same as (**b**) for the SWCNT film integrated with an engineered antenna (with bottom metal plane). (**e**) Electromagnetic field distribution of the engineered antenna integrated SWCNT film on a *y-z* cross section at *x* = 0. (**f**) *A*_j_ as a function of the thickness of the insulator spacer in the engineered antenna integrated SWCNT film for *y*-polarization at 1 THz. The junction area is defined as a 5 μm × 100 μm × 2 μm box at the center of the SWCNT film, as shown by the gray shade in the inset. *E*_0_ in (**b**,**d**) denotes the electric field of the incident Gaussian beam at the junction.

**Figure 4 sensors-21-05221-f004:**
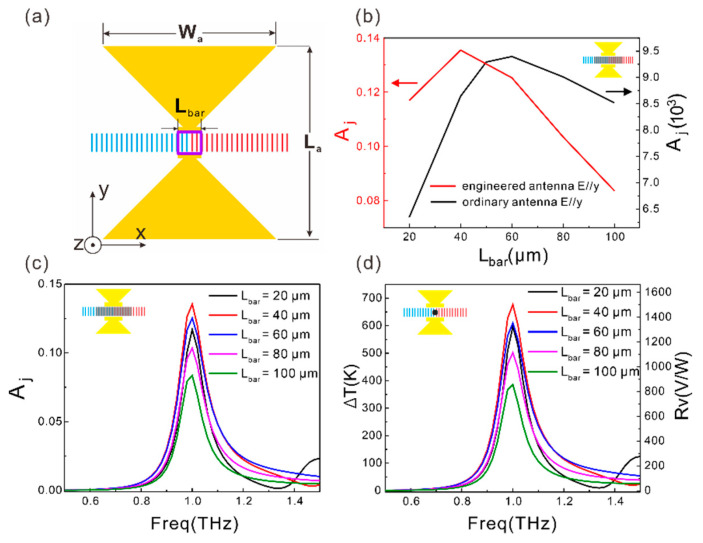
(**a**) Schematic diagram of the bowtie antennas integrated aligned SWCNT film. The purple region represents the part of the SWCNT film between the two extension bars. *R*_load_ is decided by the SWCNTs within this region. (**b**) *A*_j_ as a function of *L*_bar_ for *y*-polarized incident light at 1 THz. Black line: ordinary antenna integrated aligned SWCNT film; Red line: engineered antenna integrated aligned SWCNT film. (**c**) Spectra of *A*_j_ at different *L*_bar_ for the engineered antenna integrated SWCNT film for *y*-polarization. (**d**) Spectra of the junction temperature increase (Δ*T*) and the responsivity at different *L*_bar_ for the engineered antenna integrated SWCNT film for *y*-polarized light. when *L*_bar_ = 20, 40, 60, 80 or 100 μm, *L*_a_ = *W*_a_ = 171, 154, 133, 115 or 100 μm, respectively.

**Figure 5 sensors-21-05221-f005:**
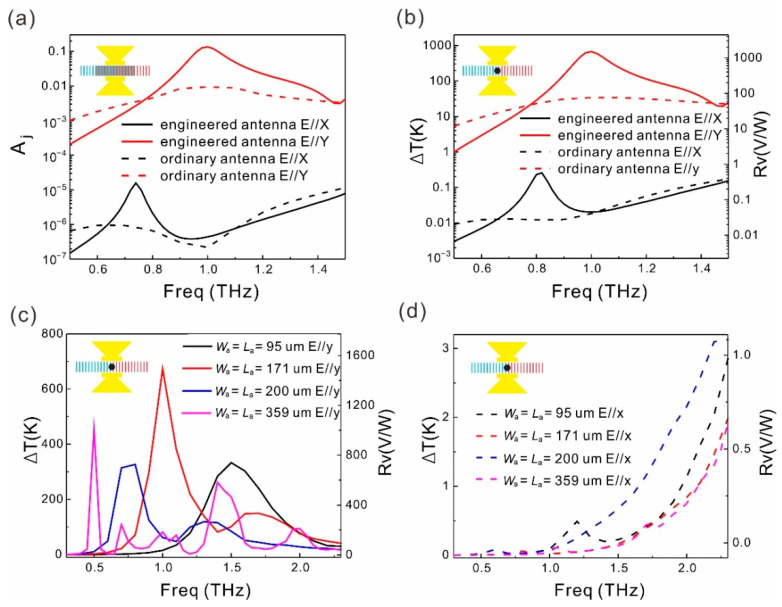
(**a**) Spectra of *A*_j_ of the ordinary antenna integrated SWCNT film and that of the engineered antenna integrated SWCNT film for *x*- and *y*-polarized incident light at 1 THz. (**b**) Spectra of Δ*T* and *R*_V_ of the ordinary antenna integrated SWCNT film and the engineered antenna integrated SWCNT film for *x*- and *y*-polarized incident light at 1 THz. (**c**,**d**) Spectra of Δ*T* and *R*_V_ of the engineered antenna integrated SWCNT film for four different antenna sizes (*L*_a_ = *W*_a_ = 95 μm, 150 μm, 200 μm, 379 μm) excited by *y*- and *x*-polarized light, respectively.

**Table 1 sensors-21-05221-t001:** Peak Rv, peak PER, and enhancement times of Rv and PER compared to that of the ordinary antenna integrated devices.

	Insulator Thinkness	Peak *R*v	Peak PER	Enhancement of Peak *R*v	Enhancement of Peak PER
Engineered antenna *L*_a_ = *W*_a_ = 95 μm	17 μm	735.5 V/W @ 1.5 THz	1481.1 @ 1.5 THz	8.1 @ 1.5 THz	2.7 @ 1.5 THz
Engineered antenna *L*_a_ = *W*_a_ = 171 μm	13 μm	1500 V/W @ 1 THz	33400 @ 1 THz	19.9 @ 1 THz	19.1 @ 1 THz
Engineered antenna *L*_a_ = *W*_a_ = 200 μm	32 μm	693.7 V/W @ 0.7 THz	14732.8 @ 0.7 THz	9.3 @ 0.7 THz	9.4 @ 0.7 THz
Engineered antenna *L*_a_ = *W*_a_ = 379 μm	13 μm	1004.9 V/W @ 0.5 THz	3.04 × 10^5^ @ 0.5 THz	15.7 @ 0.5 THz	22.3 @ 0.5 THz

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
