# Peer review of "Carbon Nanotube Far Infrared Detectors with High Responsivity and Superior Polarization Selectivity Based on Engineered Optical Antennas"

_sensors, 2021, doi:10.3390/s21155221_

Round 1

Reviewer 1 Report

Comments to the authors:

I suggest improvements.

  • Page 4: lines 147 – 149: What is relation between antenna dimensions and resonant frequency 1 THz, and why is this important?
  • Figures 2, 3, and 4 – Are there any experimental results or this are just simulations?
  • Page 7: line 234: Could authors comment advantages and disadvantages of the peak responsivity shift from 1.5 THz to 0.5 THz, and why is this important?

Author Response

Dear reviewer, I will reply you with illustrations and formulas in the attachment.

Thank you very much for reviewing our manuscript “Carbon nanotube far infrared detectors with high responsivity and superior polarization selectivity based on engineered optical antennas” (ID: sensors-1290128) and providing constructive suggestions on that. Below is our reply to the questions and comments. After that, the corresponding changes to the manuscript are listed in the following.

  1. Page 4: lines 147 – 149: What is relation between antenna dimensions and resonant frequency 1 THz, and why is this important?

Thank you for this insightful question! The ordinary bowtie antenna as shown in Fig. 2 (a) in the manuscript can be regarded as an electric dipole antenna with an extended bandwidth [1]. The resonance frequency f0 is decided by the length of the antenna La like f0 = c/2La. c is the light speed. In our case, in order for the resonance at 1 THz, La is set to be 150 mm. Concerning the engineered bowtie antenna as we proposed in this work (Fig. 2 (b) and Fig. 3 (c) in the manuscript), each wing of the bowtie antenna couples with its image due to the deep subwavelength thickness of the dielectric spacer and then forms a magnetic dipole, as exhibited in Fig. R1 (a) and (b). At the magnetic dipole resonance, the incident light is efficiently coupled into the system, leading to an enhanced local field that significantly improves the responsivity of the carbon nanotube detector. Since the magnetic dipole is less radiative than the electric dipole, the Q factor of the engineered bowtie antenna resonance is 4.6 times higher than that of the ordinary bowtie antenna, as shown in Fig. 5 (a) and (b) in the manuscript.

Fig. R1: (a) Schematic diagram of the engineered bowtie antenna coupling to the incident resonance wave. Each wing of the bowtie antenna couples with its image and then forms a magnetic dipole. (b) The distribution field diagram of the engineered bowtie antenna in the y-z section, x=0. The surface color aberration represents the modulus of the magnetic field, and the arrows represents the electric field vector (Ey, Ez). The results in (b) are in line with the coupling model established in (a). (c) Schematic diagram of one side of the bowtie antenna..

The resonant frequency of the engineered bowtie antenna can be estimated by a patch antenna model [2-4]:

where  represents effective dielectric constant of the insulator layer,  the relative dielectric constant of the insulator layer, h the height of dielectric substrate,  the resonant frequency, W the width of the antenna, Wc the width of the extension bar, and L the length of each wing of the antenna. Concerning the device presented in Fig. R1, same as that in Fig. 2 (b) and Fig. 3 (c) in the manuscript, , h = 13 mm. The resonant frequency  is calculated to be 1.08 THz. In addition, since the two wings are close to each other, the coupling between the two magnetic dipole resonator red-shifts the resonance a little bit to 1 THz.

There is no special reason to set the resonance at 1 THz. When the resonance is shifted to other frequencies as shown in Fig. 5 (c) and (d) in the manuscript, the conclusion is similar.

The corresponding discussions are added to manuscript. And the Fig. 3 is modified accordingly.

  1. Figures 2, 3, and 4 – Are there any experimental results or this are just simulations?

The results in Figures 2, 3, and 4 are all produced by simulation. Infrared detectors based on aligned SWCNT films with diverse doping have been reported by many previous works, such as Nano Lett. 14, 3953 (2014) and ACS Nano 7, 7271 (2013), and have been regarded as a promising candidate for far infrared detectors. In this work, we propose to integrate an engineered bowtie antenna with an aligned SWCNT film based infrared detector and theoretically prove that both the responsivity and the polarization selectivity can be enhanced by more than 2 orders of magnitudes, which is an encouraging progress for the technical route of CNT based infrared detection. Our study is based on the simulations of both the optical field and the thermal field, so the results are more accurate than simulations of only the optical field.

  1. Page 7: line 234: Could authors comment advantages and disadvantages of the peak responsivity shift from 1.5 THz to 0.5 THz, and why is this important?

The peak responsivity shift is intentionally implemented by changing the size of the antenna. In the absence of the antenna, the photoresponse of the aligned SWCNT film based infrared detector is broadband. Although the integration of the antenna significantly enhances the responsivity, the bandwidth is narrowed. A possible solution is to tuning the resonance of the antenna to the requested frequency for a specific application. The discussion on page 7 and the Fig. 5 (c) and (d) are used to prove that the peak responsivity can be feasibly tuned by changing the antenna size.

The following is our change list that includes responses to your suggestions in the manuscript, as well as changes to English grammar:

Change List:

Manuscirpt:

  1. Page 2 Line 73-75: the definition of RV is
  2. Page 4 Line 126: supplementary information for the Gaussian beam is added.
  3. Page 4 Line 136-144: a discussion about the intrinsic optical properties of aligned SWCNT film with limited SWCNT lengths is added.
  4. Page 4 Line 144-145: the first sentence is changed into " In order to increase the responsivity and the polarization extinction ratio of the aligned SWCNT film based detector, it is natural to add an antenna to concentrate the incident light at the junction region of the SWCNT film, as shown in Fig. 2 (a). " is added.
  5. Page 4 Line 157: the sentence is changed into " which is more than 3 orders of magnitude higher than that of the device without any antenna. " is added.
  6. Page 4 Line 159-166: a discussion about the photonic mode of the engineered antenna integrated device and the relation between the antenna dimensions and resonant frequency is added.
  7. Page 6 Line 204-205: the sentence " Each wing couples with its image and induces a magnetic dipole. " is added.
  8. Page 6 Line 200: Fig. 3 is modified to illustrate the magnetic dipole resonance mode of the engineered antenna integrated device more explicitly.
  9. Page 6 Line 206-207: the sentence is changed into “Fig. 3(e) Electromagnetic field distribution of the engineered antenna integrated SWCNT film on a y-z cross section at x=0.” is added.
  10. Page 7 Line 243-244: detailed dimensions of the antennas for different extention bar lengths are provided.
  11. Page 7 Line 256-257: the sentence is changed into " The peak Rv, peak PER, and the enhancement times of these two quantities compared to that of the ordinary antenna integrated devices are presented in Table 1. " is added.
  12. Page 8 Line 262-264: Fig. 5 (c) is splitted into two pannels.
  13. Page 8 Line 265-266: the sentence is changed into " peak Rv, peak PER, and enhancement times of Rv and PER compared to that of the ordinary antenna integrated devices. " is added.

References

  1. J. D. Kraus and R. J. Marhefka, “Antennas: For All Applications”, Publishing House of Electronics Industry.
  2. T. Nahar and O. P. Sharma, " A Modified Multiband Bow Tie Antenna Array used for L band Application," International Journal of Engineering Research & Technology (IJERT), ISSN: 2278-0181.
  3. Chen Wen-jun, Li Bin-hong and Xie Tao, " A Resonant Frequency Formula of Bow-tie Antenna and Its Application," Antennas and Propagation Society International Symposium, 2004. IEEE.

Reviewer 2 Report

The authors report on simulations to aid in the optimization of the engineering of far infrared detectors based on aligned arrays of single-walled carbon nanotubes (SWCNT). This concise report is well written and demonstrates the potential for considerable enhancement of detector performance based on their optimization. I would like the authors to address three minor points before acceptance for publication.

1) How is the intrinsic anisotropy and polarization selectivity of the sensor array influenced by the aspect ratio of the SWCNTs? Please address also that, while the widths of the SWCNT are in the deep-subwavelength scale, the lengths are usually not. Does this need to be considered in the analysis?

2) RV is not clearly defined in the text.

3) Figure 5c is too crowed. Split into two panels so that the data is more distinct. Then present two panels above and two panels below in two rows rather than just one row.

Author Response

Dear reviewer, I will reply you with illustrations and formulas in the attachment.

Thank you very much for reviewing our manuscript “Carbon nanotube far infrared detectors with high responsivity and superior polarization selectivity based on engineered optical antennas” (ID: sensors-1290128) and providing constructive suggestions on that. Below is our reply to the questions and comments. After that, the corresponding changes to the manuscript are listed in the following.

  1. How is the intrinsic anisotropy and polarization selectivity of the sensor array influenced by the aspect ratio of the SWCNTs? Please address also that, while the widths of the SWCNT are in the deep-subwavelength scale, the lengths are usually not. Does this need to be considered in the analysis?

Thank you for this insightful question! In fact, since the real part of the parallel permittivity of the aligned SWCNT film (Re(e||)) is negative in the frequency range from 0 to 2.8 THz (Fig. R1 (a)), SWCNTs with limited length can support localized surface plasmon modes [1]. The diagram of the electric field distribution in Fig. R1 (b) shows that this mode is a fundamental LSPR. By changing the length of the SWCNTs, the LSPR can be shifted in the frequency range as shown in Fig. R1 (c). When Wb = 5 mm, the resonant frequency is 1.77 THz. When Wb = 20 mm, the resonant frequency is shifted to 1 THz. Although the LSPR of the aligned SWCNT film could also enhance the light absorption, the light coupling efficiency is low and the light field is not concentrated at the junction region. So the temperature rise at the junction and the responsivity enhancement is much smaller than those induced by the optical antenna. Fig. R1 (c) shows that the LSPR frequency changes with the Wb of SWCNT. For tWb = 5 mm, the SWCNT film without any antenna has an RV of 3.36 V/W at 1 THz (Fig. R1 (c)), which is 443.9 times lower than that induced by the engineered bowtie antenna. The polarization extinction ratio (PER) of the SWCNT film without any antenna is 41.9, which is 793.5 times lower that of the engineered antenna integrated device. When the LSPR is shifted to 1 THz by increasing the SWCNT length to 20 mm, the RV at 1 THz is increased to 25.1 V/W (Fig. R1 (c)), but still 59.3 times lower than the RV induced by the engineered antenna. And the PER is increased to 262, which is 127 times lower that of the engineered antenna integrated device.  Therefore, the aspect ratio of the length of the SWCNTs does have some influence on the device performance based on LSPR, but the effect is much less prominent than that induced by the antenna.

Fig. R1: (a) Relative permittivity of the aligned SWCNT film regarded as a uniaxial effective medium [1]. (b) Electric field distributions (|E|/|E0|) on the (x-y, z = 1 μm) and (y-z, x=0) cross section. (c) Spectra of junction temperature increase (DT) and responsivity (RV) of two devices with different SWCNT lengths (5 mm and 20 mm) for the x- and the y-polarized wave excitation. The other dimensions of the SWCNT film remain the same: 1 mm long in the x-direction and 2 μm thick, the bottom layer of the detector is 13 μm thick Teflon and the incident light is assumed to be a Gaussian beam focused on the SWCNT belt with a beam waist around 800 mm.

  1. RV is not clearly defined in the text.

RV is a shorthand for Responsivity. In this work, the responsivity is defined as the photovoltage divided by the optical power incident on the junction area (a 100 mm × 5 mm region at the p-n junction, as marked in the inset of Fig. 4 (c) and Fig. 5 (a) in the manuscript).

  1. Figure 5c is too crowed. Split into two panels so that the data is more distinct. Then present two panels above and two panels below in two rows rather than just one row.

Thanks for your advice. We have split figure 5c into two figures (fig. 5c and fig.5d) and show the spectra of ΔT and RV excited by y- and x- polarized light separately for a clearer display.

The following is our change list that includes responses to your suggestions in the manuscript, as well as changes to English grammar:

Change List:

Manuscirpt:

  1. Page 2 Line 73-75: the definition of RV is
  2. Page 4 Line 126: supplementary information for the Gaussian beam is added.
  3. Page 4 Line 136-144: a discussion about the intrinsic optical properties of aligned SWCNT film with limited SWCNT lengths is added.
  4. Page 4 Line 144-145: the first sentence is changed into " In order to increase the responsivity and the polarization extinction ratio of the aligned SWCNT film based detector, it is natural to add an antenna to concentrate the incident light at the junction region of the SWCNT film, as shown in Fig. 2 (a). " is added.
  5. Page 4 Line 157: the sentence is changed into " which is more than 3 orders of magnitude higher than that of the device without any antenna. " is added.
  6. Page 4 Line 159-166: a discussion about the photonic mode of the engineered antenna integrated device and the relation between the antenna dimensions and resonant frequency is added.
  7. Page 6 Line 204-205: the sentence " Each wing couples with its image and induces a magnetic dipole. " is added.
  8. Page 6 Line 200: Fig. 3 is modified to illustrate the magnetic dipole resonance mode of the engineered antenna integrated device more explicitly.
  9. Page 6 Line 206-207: the sentence is changed into “Fig. 3(e) Electromagnetic field distribution of the engineered antenna integrated SWCNT film on a y-z cross section at x=0.” is added.
  10. Page 7 Line 243-244: detailed dimensions of the antennas for different extention bar lengths are provided.
  11. Page 7 Line 256-257: the sentence is changed into " The peak Rv, peak PER, and the enhancement times of these two quantities compared to that of the ordinary antenna integrated devices are presented in Table 1. " is added.
  12. Page 8 Line 262-264: Fig. 5 (c) is splitted into two pannels.
  13. Page 8 Line 265-266: the sentence is changed into " peak Rv, peak PER, and enhancement times of Rv and PER compared to that of the ordinary antenna integrated devices. " is added.

References

[1]:   B. Chen, Z. Ji, J. Zhou, Y. Yu, X. Dai, M. Lan, Y. Bu, T. Zhu, Z. Li, J. Hao, and X. Chen, "Highly polarization-sensitive far infrared detector based on an optical antenna integrated aligned carbon nanotube film," Nanoscale 12(22), 11808–11817 (2020).
